# Anti-Inflammatory and Protein Tyrosine Phosphatase 1B Inhibitory Metabolites from the Antarctic Marine-Derived Fungal Strain *Penicillium glabrum* SF-7123

**DOI:** 10.3390/md18050247

**Published:** 2020-05-09

**Authors:** Tran Minh Ha, Dong-Cheol Kim, Jae Hak Sohn, Joung Han Yim, Hyuncheol Oh

**Affiliations:** 1Institute of Pharmaceutical Research and Development, College of Pharmacy, Wonkwang University, Iksan 54538, Korea; minhha19@outlook.com (T.M.H.); kimman07@hanmail.net (D.-C.K.); 2College of Medical and Life Sciences, Silla University, Busan 46958, Korea; jhsohn@silla.ac.kr; 3Korea Polar Research Institute, KORDI, 7-50 Songdo-dong, Yeonsu-gu, Incheon 21990, Korea

**Keywords:** marine-derived fungi, anti-inflammation, anti-neuroinflammation, PTP1B

## Abstract

A chemical investigation of the marine-derived fungal strain *Penicillium glabrum* (SF-7123) revealed a new citromycetin (polyketide) derivative (**1**) and four known secondary fungal metabolites, i.e, neuchromenin (**2**), asterric acid (**3**), myxotrichin C (**4**), and deoxyfunicone (**5**). The structures of these metabolites were identified primarily by extensive analysis of their spectroscopic data, including NMR and MS data. Results from the initial screening of anti-inflammatory effects showed that **2**, **4**, and **5** possessed inhibitory activity against the excessive production of nitric oxide (NO) in lipopolysaccharide (LPS)-stimulated BV2 microglial cells, with IC_50_ values of 2.7 µM, 28.1 µM, and 10.6 µM, respectively. Compounds **2**, **4**, and **5** also inhibited the excessive production of NO, with IC_50_ values of 4.7 µM, 41.5 µM, and 40.1 µM, respectively, in LPS-stimulated RAW264.7 macrophage cells. In addition, these compounds inhibited LPS-induced overproduction of prostaglandin E_2_ in both cellular models. Further investigation of the most active compound (**2**) revealed that these anti-inflammatory effects were associated with a suppressive effect on the over-expression of inducible nitric oxide synthase and cyclooxygenase-2. Finally, we showed that the anti-inflammatory effects of compound **2** were mediated via the downregulation of inflammation-related pathways such as those dependent on nuclear factor kappa B and p38 mitogen-activated protein kinase in LPS-stimulated BV2 and RAW264.7 cells. In the evaluation of the inhibitory effects of the isolated compounds on protein tyrosine phosphate 1B (PTP1B) activity, compound **4** was identified as a noncompetitive inhibitor of PTP1B, with an IC_50_ value of 19.2 µM, and compound **5** was shown to inhibit the activity of PTP1B, with an IC_50_ value of 24.3 µM, by binding to the active site of the enzyme. Taken together, this study demonstrates the potential value of marine-derived fungal isolates as a bioresource for bioactive compounds.

## 1. Introduction

Marine-derived fungi have been suggested as a unique source of bioactive secondary metabolites [1,2]. In recent years, research interest in marine-derived fungi as a valuable resource of bioactive compounds has significantly increased, resulting in the exploration of a variety of novel metabolites, consisting of polyketides (40%), alkaloids (20%), peptides (15%), terpenoids (15%), prenylated polyketides (7%), as well as shikimates (2%) and lipids (1%) [3]. The chemical study of the ethyl acetate (EtOAc) extract obtained from the culture of the marine-derived fungal strain *Penicillium glabrum* (SF-7123) has been the focus of our continuing efforts to find bioactive secondary metabolites from marine-derived fungal strains collected in Antarctic area. The fungal strain SF-7123 was cultured on Petri agar plates (containing 3% NaCl) at 25 °C for three weeks. The culture media were extracted with EtOAc, and the filtered extracts were then concentrated in vacuo to provide a crude extract. A combination of chromatographic methods was employed to yield a new citromycetin derivative (**1**) and four known secondary metabolites (**2**–**5**) from the crude extract (Figure 1).

The chemical investigation of the extract from the fungal strain SF-7123 utilized a bioassay system for the evaluation of anti-inflammatory effects in cellular models to detect the bioactive fractions and pure fungal metabolites from the crude extract. Inflammation is recognized as a vital reaction of the body to injury or infection. Macrophages and microglia (the resident macrophage-like cells of the central nervous system) have been repeatedly reported to exert a key role in the immune system [4,5]. They are activated in response to various stimuli and lead to phagocytosis of damaged macrophages and neuronal cells to protect tissues and prevent damage to the brain and body. However, their sustained activation leads to the release of pro-inflammatory mediators such as inducible nitric oxide synthase (iNOS), cyclooxygenase-2 (COX-2), NO, and prostaglandin E_2_ (PGE_2_), as well as of pro-inflammatory cytokines, including tumor necrosis factor (TNF)-α, interleukin (IL)-1β, IL-6, and IL-12. These mediators and cytokines are known to be detrimental to cells or tissues and to cause various inflammatory diseases, including inflammatory bowel disease, Alzheimer’s disease, Parkinson’s disease, and multiple sclerosis [6,7,8,9]. Therefore, controlling the production of pro-inflammatory mediators/cytokines could be regarded as a reasonable target for the prevention and/or treatment of inflammatory diseases. For many years, the RAW264.7 cell line (Abelson murine leukemia virus-transformed macrophage cells derived from male BALB/c mice) has been widely accepted as an in vitro model to investigate cellular inflammation responses. Likewise, BV2 cells (raf/myc-immortalized murine microglia) have been frequently employed to model the reactions of microglia in vivo.

To gain further information with respect to the biological effects of the fungal metabolites isolated in this study, an enzymatic assay system to evaluate the inhibitory effects on protein tyrosine phosphatase 1B (PTP1B) was also employed throughout the study. PTP1B is a major negative regulator of insulin and leptin signaling pathways. It has been reported that PTP1B is related to inflammation in many tissues, such as hypothalamus and lungs, and plays an important role in lipopolysaccharide (LPS)-induced activation of microglia and macrophages [10,11]. Furthermore, many studies have indicated that PTP1B expression is increased under many pathophysiological conditions such as inflammation, cancer, and diabetes, suggesting novel therapeutic implications for these PTP1B inhibitors in the treatment of such diseases [12].

## 2. Results and Discussion

### 2.1. Isolation and Structure Determination of Compounds ***1***–***5***

Compound 1 was isolated as an amorphous powder, optically active, with an ${\left[\alpha \right]}_{D}^{20}$ value of −344.63 (c 0.48, CH_3_OH). The HRESIMS (Appendix A) of **1** showed an ion peak at m/z 263.0911 [M + H]^+^ (calcd for C_14_H_1__5_O_5_, 263.0919), determining the molecular formula C_14_H_14_O_5_. The ^1^H NMR spectrum of **1** (Table 1, Appendix A) disclosed the typical signals for two aromatic protons at δ_H_ 6.34 (1H, s, H-7) and 7.06 (1H, s, H-10), a singlet peak at δ_H_ 3.83 (3H, H-12) for methoxy protons, and a signal for methyl protons at δ_H_ 1.55 (3H, d, *J* = 6.4, H-11). In addition, there were three downfield-shifted signals corresponding to an oxymethine proton at δ_H_ 4.71 (1H, m, H-2) and oxymethylene protons at δ_H_ 4.73 (1H, d, *J* = 12.0, H-5a) and δ_H_ 5.03 (1H, d, *J* = 12.0, H-5b). The remaining signal at δ_H_ 2.53 (2H, m, H-3) was assigned to a methylene proton. The ^13^C NMR spectra (Appendix A) of **1** displayed 14 carbons, including 9 sp^2^ signals assigned to a carbonyl group at δ_C_ 191.0 (C-4),and two aromatic methine carbons at δ_C_ 104.8 (C-7) and δ_C_ 108.9 (C-10). In addition, six sp^2^ quaternary carbon signals at δ_C_ 102.4 (C-4a), 156.3 (C-6a), 154.5 (C-8), 144.8 (C-9), 107.9 (C-10a), and 165.7 (C-10b) were present in the spectrum. The two oxygenated carbons at δ_C_ 77.3 and δ_C_ 64.1 were identified as a methine (C-2) and a methylene (C-5), respectively. The ^13^C NMR spectra of **1** also displayed signals for a methoxy group at δ_C_ 57.0 (C-12), a methylene carbon at δ_C_ 43.5 (C-3), and a methyl group at δ_C_ 20.6 (C-11). A literature review revealed that the ^1^H and ^13^C NMR data of **1** were similar to those reported for neuchromenin [13,14], indicating close structural similarity between the two compounds, except for the presence of a methoxy group in **1**. A detailed analysis of the COSY and HMBC spectra of **1** (Table 1, Appendix A) confirmed the planar structure of **1** as a derivative of neuchromenin with the replacement of the hydroxy group by a methoxy group attached to C-9. This was supported by the observation of an HMBC correlation from 9-OCH_3_ to C-9. Consequently, on the basis of this evidence, the structure of compound **1** was determined as shown in Figure 1 and named 9-*O*-methylneuchromenin.

The NMR data of compound **2** (Appendix A) were almost identical to those of **1**, indicating a pyranchromene skeleton typical of citromycetin analogues. A close inspection of the ^1^H and ^13^C NMR data of compound **2** and their comparison with those found in the literature eventually led to the assignment of the structure of neuchromenin, which has been isolated from the culture broth of *Eupenicillium javanicum* var*. meloforme* PF1181. Neuchromenin reportedly induced neurite outgrowth in PC12 pheochromocytoma cells in a rat model [13]. The absolute configuration of naturally occurring neuchromenin was previously determined to be *S*-configuration via the synthetic study of the enantiomers of neuchromenin [14]. Close similarities with data reported in the literature are found for the NMR data and the well-matched specific rotation value of **2** [15], suggesting that compound **2** must have the same absolute configuration at C-2 as that of (−)-neuchromenin. Accordingly, the absolute configuration of **1** was suggested to be analogous to that of **2**, because these fungal metabolites were produced by the same fungal strain.

The structures of the remaining compounds were also determined based on the analysis of their NMR (Appendix A) and MS data, along with a comparison with previously published data in the literature. A diphenyl ether was isolated in this study, which was identified as asterric acid (**3**), known as an endothelin binding inhibitor [16,17]. Compound **4** was identified as a citromycetin derivative, i.e., myxotrichin C [18]. There are no previous reports on the biological effects of this metabolite. The structure of compound **5** was elucidated and found to be deoxyfunicone, which could be classified as a phenolic polyketide and a funicone analogue [19]; it was described as an HIV-1-integrase inhibitor [20].

### 2.2. Effects of Secondary Metabolites Isolated from SF-7123 on the Production of Pro-Inflammatory Mediators

LPS is a major cell wall component of Gram-negative bacteria. LPS is an intensive stimulator of macrophages and microglia, and thus treatment of these cells with LPS is frequently employed in research related to the inflammatory phenomenon [21,22]. In the current study, all the isolated compounds (**1**–**5**) were investigated for their anti-neuroinflammatory and anti-inflammatory effects by examining their effects on the overproduction of NO and PGE_2_ in LPS-stimulated BV2 microglia and RAW264.7 macrophage cells. LPS stimulation increased the production of NO and PGE_2_ in both cell lines. Pre-treatment with **2**, **4**, and **5** attenuated these responses, with different IC_50_ values as shown in Table 2. Compounds **1** and **3** did not significantly affect to the alteration of NO production at 80.0 μM. On the basis of the IC_50_ values of **1** and **2** and their respective structures, we reasoned that the replacement of the hydroxy group at position C-9 in neuchromenin (**2**) by a methoxy group would decrease this inhibitory effect significantly. The most active compound in the assay was identified to be **2**. Thus, this compound was selected to further examine the anti-inflammatory effect and to reveal its underlying mechanisms. Compound **2** effectively decreased the mRNA over-expressions of IL-1β, TNF-α, IL-6, and IL-12 in LPS-stimulated BV2 and RAW264.7 cells (Figure 2) when the cells were activated with LPS (1 µg/mL) for 12 h. Moreover, compound **2** attenuated the protein expression of iNOS and COX-2 in a dose-dependent manner in LPS-stimulated BV2 cells (Figure 3A). In LPS-stimulated RAW264.7 cells, compound **2** dose-dependently attenuated the protein expression of COX-2, and a significant attenuation of iNOS expression was observed at a concentration of 4.0 μM (Figure 3B). To confirm that the observed anti-inflammatory effects did not arise from their cytotoxicity, the MTT assay was conducted in the presence of the compounds tested, and the results showed that compounds **1**–**5** were not cytotoxic to these cells at doses up to 80.0 μM (data not shown).

### 2.3. Effect of Compound ***2*** on NF-κB and Mitogen-Activated Protein Kinase (MAPK) Pathways 

Nuclear factor-kappa B (NF-κB) and MAPK are essential for the regulation of pro-inflammatory mediators and pro-inflammatory cytokines. Therefore, the pathways they regulate play a vital role in the inflammatory reaction. NF-κB is a transcription factor normally kept in the cytoplasm by inhibitor kappa B-α (IκB-α). Inflammatory stimulants such as LPS induce IκBs phosphorylation, which leads to the release of an NF-κB dimer (p50 and p65) that translocate into the nucleus, resulting in the transcription of inflammatory genes such iNOS, COX-2, NO, PGE_2_, TNF-α, IL-1β, IL-6, and IL-12 [23,24,25]. Therefore, we examined the effects of compound **2** on the LPS-stimulated upregulation of the NF-κB pathway in BV2 and RAW264.7 cells. Immunoblotting assays revealed that compound **2** suppressed the activation of the NF-κB pathway in cells that were stimulated with LPS. For example, compound **2** reversed the phosphorylation and degradation of IκB-α (Figure 4A,B) and blocked the translocation of the NF-κB dimer (p50 and p65) into the nucleus (Figure 4C–F). Furthermore, compound **2** decreased the DNA binding activity of the p65 subunit (Figure 4G,H) in LPS-stimulated BV2 and RAW264.7 cells.

MAPKs have been reported to play a critical role in a variety of cellular conditions such as cell death, cell growth, differentiation, proliferation, and immune responses [26,27]. MAPKs consist of three major signaling kinases, i.e., extracellular signal-regulated kinases 1 and 2 (ERK1/ERK2), c-Jun N-terminal kinases (JNKs), and p38 MAPK. Specifically, p38 MAPK is one of the MAPKs that regulate the inflammatory responses and is considered a therapeutic target for anti-inflammatory treatments. In this investigation, we examined the effects of compound **2** on LPS-stimulated activation of MAPKs in BV2 and RAW264.7 cells. When the cells were stimulated with LPS for 1 h, the phosphorylation levels of MAPKs were significantly upregulated. However, pre-treatment with compound **2** inhibited LPS-induced p38 MAPK phosphorylation in BV2 and RAW264.7 cells, whereas the increased phosphorylation levels of ERK and JNK MAPKs were unchanged (Figure 5). On the basis of these results, it was postulated that compound **2** exhibited anti-inflammatory and anti-neuroinflammatory effects via the suppression of the activation of NF-κB and p38 MAPK pathways. 

### 2.4. PTP1B Inhibitory Effects of the Isolated Metabolites ***1***–***5***

In addition to the investigation of the anti-inflammatory and anti-neuroinflammatory effects of the isolated metabolites in cellular models, this study attempted to assess the inhibitory effect of the isolated compounds against PTP1B activity. In this enzymatic assay, a known phosphatase inhibitor, ursolic acid, was used as a positive control, and *p*-nitrophenol phosphate (*p*NPP) was used as an enzyme substrate [28,29]. When compounds **1**–**5** at 50 µM concentration were incubated with PTP1B in the presence of the substrate, compounds **4** and **5** were shown to significantly inhibit the activity of PTP1B, whereas the inhibitory activity of compounds **1**–**3** was less pronounced. Furthermore, the inhibitory effects of compounds **4** and **5** were concentration-dependent, and their IC_50_ values were determined to be 19.2 µM and 24.3 µM, respectively (Table 3).

Next, the effects of compounds **4** and **5** on the kinetic profile of PTP1B -atalyzed *p*NPP hydrolysis were analyzed, as described in the methods section. When the enzyme assay was performed for the *p*NPP substrate in the presence or absence of compound **4** at different concentrations, the enzyme *K*_m_ did not change with increasing inhibitor concentration, whereas *V*_max_ decreased, as depicted in a Lineweaver–Burk plot in Figure 6A. Therefore, compound **4** was determined to be a noncompetitive inhibitor, suggesting that the compound may bind to an allosteric site within PTP1B or to an enzyme–substrate complex. Furthermore, the kinetic analysis revealed that the inhibition mode of compound **5** was competitive, as the Lineweaver–Burk plot showed an increase of the *K*_m_, without changes in the *V*_max_ value (Figure 6B). This results indicate that compound **5** might bind to the active site of PTP1B.

## 3. Experimental

### 3.1. General Experimental Procedures

Optical rotations were recorded using a Jasco P-2000 digital polarimeter. NMR spectra (1D and 2D) were recorded in CD_3_OD or CDCl_3_ using a JEOL JNM ECP-400 spectrometer (400 MHz for ^1^H and 100 MHz for ^13^C) using standard JEOL pulse sequences. The chemical shifts were referenced relative to the respective residual solvent signals (CD_3_OD: *δ*_H_ 3.30/*δ*_C_ 49.0 and CDCl_3_: *δ*_H_ 7.26/*δ*_C_ 77.0). HMQC and HMBC experiments were performed using an optimized sequence for ^1^*J*_CH_ = 140 Hz and ^n^*J*_CH_ = 8 Hz, respectively. HRESIMS data were obtained using an ESI Q-TOF MS/MS system (AB SCIEX Triple). HPLC separations were performed on a prep-C_18_ column (21.2 × 150 mm; 5 µm particle size) at a flow rate of 5 mL/min. UV detection at 210 nm and 254 nm was utilized. TLC analysis was performed on Kieselgel 60 F_254_ or RP-18 F_254s_ plates. Flash Column chromatography was conducted using YMC octadecyl-functionalized silica gel (C_18,_ 75 µm).

### 3.2. Fungal Material and Fermentation 

*P. glabrum* SF-7123 was isolated from sediments that were collected using a dredge at the Ross Sea (77°34.397′ N, 166°10.865′ W) on 10 January 2015. One gram of the sample was mixed with sterile seawater (10 mL), and a portion (0.1 mL) of the sample was processed according to the spread plate method in potato dextrose agar (PDA) medium containing sterile seawater. The plate was incubated at 25 °C for 14 days. The isolates were cultured several times to obtain a final pure culture, and selected cultures were preserved at −70 °C. The identification of the fungal strain SF-7123 was conducted by the analysis of the 28S ribosomal RNA (rRNA) gene sequence. A GenBank search with the 28S rRNA gene of SF-7123 (GenBank accession number KY563089) indicated *P. glabrum* (JN938946), *Penicillum spinulosum* (HM469405), and *Penicillum multicolor* (HM469407) as the closest matches, showing sequence identities of 100%, 99.64%, and 97.13%, respectively. Therefore, the marine-derived fungal strain SF-7123 was characterized as *P. glabrum*.

### 3.3. Extraction and Isolation of Compounds ***1***–***5***

The fungal strain SF-7123 was cultured in 10 Fernbach flasks. Each flask contained 300 mL of PDA medium with 3% NaCl (w/v). The flasks were individually inoculated with 2 mL of seed cultures of the fungal strain and incubated at 25 °C for 14 days. The fermented culture media were combined and extracted with EtOAc (4 L). The combined EtOAc extracts were then filtered through filter paper and evaporated to dryness, resulting in a crude extract of SF7123 (1.0 g). The crude extract was fractionated by reversed-phase (RP) C_18_ flash column chromatography (4.5 × 30 cm), eluting with a stepwise gradient of 20%, 30%, 40%, 50%, 60%, 80%, and 100% (v/v) MeOH in H_2_O (400 mL each) to provide seven subfractions, SF7123-1 to SF7123-8. Fraction SF7123-4 was subjected to column chromatography (3 × 35 cm) using Sephadex LH-20 as the stationary phase and a 3/1 (v/v) mixture of MeOH in water as the mobile phase to provide metabolite **4** (2.5 mg) and two sub-fractions named SF7123-4-1 and SF7123-4-2. Subfraction SF7123-4-2 was then chromatographed using an RP C_18_ column (1.2 × 30 cm) and eluted with MeOH in water [2/3 (v/v)] to create three subfractions. Among these subfractions, subfraction SF7123-4-2-1 was further separated on a C_18_ prep HPLC (27%–43% CH_3_CN in H_2_O (0.1% HCOOH) over 16 min) to provide metabolite **2** (6.5 mg, t_R_ = 15 min). Similarly, subfraction SF7123-4-2-2 was purified by using C_18_ prep HPLC (35%–50% CH_3_CN in H_2_O (0.1% HCOOH) over 16 min) to afford metabolite **1** (4.3 mg, t_R_ = 15.5 min). Fraction SF7123-5 was subjected to column chromatography using Sephadex LH-20 (2.0 × 30 cm). The column was subsequently eluted with a mixture of MeOH in H_2_O [3/1 (v/v)] to separate the fraction into three subfractions, i.e., SF7123-5-1 to SF7123-5-3. The first subfraction, SF7123-5-1, was subjected to C_18_ prep HPLC (48%–70% CH_3_CN in water (0.1% HCOOH) over 18 min) to afford compound **5** (6 mg, t_R_ = 17.0 min). Compound **3** (2 mg, t_R_ = 23.0 min) was isolated from the second subfraction, SF7123-5-2, by RP C_18_ prep HPLC (35%–65% CH_3_CN in H_2_O (0.1% HCOOH) over 25 min).

Compound 9-O-methylneuchromenin (1): Yellow powder; [*α*]_D_ −344 (*c* = 0.72, MeOH); ^1^H and ^13^C NMR data, see Table 1; HRESIMS *m/z*: 263.0911 [M + H]^+^ (calcd. for C_14_H_15_O_5_, 263.0919).

### 3.4. Cell Culture and Viability Assay

RAW264.7 and BV2 cells were maintained at 5 × 10^5^ cells/mL in Dulbecco’s modified Eagle’s medium (DMEM) containing 10% fetal bovine serum (FBS), penicillin G (100 U/mL), streptomycin (100 mg/L), and L-glutamine (2 mM) and incubated at 37 °C in a humidified atmosphere containing 5% CO_2_. DMEM, FBS, and other tissue culture reagents were purchased from Gibco BRL Co. Cell viability was evaluated by the MTT assay as described in our previous report [30].

### 3.5. Nitrite Determination

To determine the nitrite concentration in the medium as an indicator of NO production, the Griess reaction was carried out as described in our previous report [30].

### 3.6. Preparation of Cytosolic and Nuclear Fractions

An Affymetrix Nuclear Extraction kit (Affymetrix Inc., Santa Clara, CA, USA) was used to extract the cytosolic and nuclear fractions of the cells. The lysis of each fraction was conducted according to the manufacturer’s instructions. The details regarding the preparation of the cytosolic and nuclear fractions have been described previously [30].

### 3.7. Western Blot Analysis

Western blot analysis was carried out according to our previous report [30]. Briefly, cells were harvested by centrifugation at 200 g for 3 min, followed by a washing step with RIPA lysis buffer (25 mM Tris-HCl buffer (pH 7.6), 150 mM NaCl, 1% NP-40, 1% sodium deoxycholate, and 0.1% SDS). Primary antibodies (COX-2: sc-1745; iNOS: sc-650; IкB-α: sc-371; p-IкB-α: sc-8404; p50: sc-7178; and p65: sc-8008) were purchased from Santa Cruz Biotechnology, and secondary antibodies (mouse: ap124p; goat: ap106p; and rabbit: ap132p) were purchased from Millipore.

### 3.8. DNA-Binding Activity of NF-κB

The DNA-binding activity of NF-κB in the nuclear extracts was measured using the TransAM^®^ kit (Active Motif, Carlsbad, CA, USA); three independent assays for each sample were conducted according to the manufacturer’s instructions.

### 3.9. PGE_2_ Assay

The level of PGE_2_ present in each sample was determined using a commercially available kit from R&D Systems (Minneapolis, MN, USA); the details of this assay were described previously [30]. Three independent assays were performed according to the manufacturer’s instructions.

### 3.10. Quantitative Real-Time Reverse-Transcription PCR (qRT-PCR)

Triplicate quantitative reverse-transcription polymerase chain reaction analysis was conducted using the following primers: 5′-AAT TGG TCA TAG CCC GCA CT-3′ and 5′-AAG CAA TGT GCT GGT GCT TC-3, forward and reverse primers for IL-1β, 5′-CCA GAC CCT CAC ACT CAC AA-3′ and 5′-ACA AGG TAC AAC CCA TCG GC-3′, forward and reverse primers for TNF-α, 5′-ACT TCA CAA GTC GGA GGC TT-3′ and 5′-TGC AAG TGC ATC ATC GTT GT-3’, forward and reverse primers for IL-6, 5′-AGT GAC ATG TGG AAT GGC GT-3′ and 5′-CAG TTC AAT GGG CAG GGT CT-3, forward and reverse primers for IL-12, and 5′-ACT TTG GTA TCG TGG AAG GAC T-3′ and 5′-GTA GAG GCA GGG ATG ATG TTC T-3′, forward and reverse primers for GADPH. The optimum conditions for PCR amplification of the cDNAs were according to the manufacturer’s instructions; the details of this assay have been previously reported [30].

### 3.11. PTP1B Assay

PTP1B (human, recombinant) used in the current study was purchased from BIOMOL Research Laboratories, Inc. PTP1B enzyme activity was determined using 2 mM of *p*NPP as a substrate in 50 mM citrate buffer solution (pH 6.0, 0.1 M NaCl, 1 mM EDTA, and 1 mM dithiothreitol). The reaction mixture was incubated at 30 °C for 30 min, and the reaction was terminated by the addition of 1 N NaOH. The amount of *p*-nitrophenol produced by the enzyme was estimated by measuring the increase in absorbance at 405 nm. The nonenzymatic hydrolysis of 2 mM *p*NPP was corrected by measuring the increase in absorbance at 405 nm in the absence of PTP1B [31]. The kinetic analysis involved the following: the assays were performed using a reaction mixture that contained different concentrations of *p*NPP (0.25 mM, 0.5 mM, 1.0 mM, and 2.0 mM) as a PTP1B substrate in the absence or presence of compounds **4** and **5**. The Michaelis–Menten constant (*K*_m_) and maximum velocity (*V*_max_) of PTP1B were determined by the Lineweaver–Burk plot using the GraphPad Prism^®^ 4 program (GraphPad Software Inc., San Diego, CA, USA).

### 3.12. Statistical Analysis

Each experiment was performed at least three times independently, and the resulting data are presented as the mean ± standard deviation. The comparison of three or more groups utilized one-way analysis of variance (ANOVA), followed by Tukey’s multiple comparison tests. Statistical analysis was performed using GraphPad Prism software, version 3.03 (GraphPad Software Inc, GraphPad Software Inc., San Diego, CA, USA). 

## 4. Conclusions

In summary, our chemical investigation of the marine-derived fungal isolate *P. glabrum* SF-7123 resulted in the isolation and identification of five secondary metabolites, including one new fungal metabolite named 9-*O*-methylneuchromenin (**1**). Although several biological activities of some known metabolites described in this study have already been reported, their anti-inflammatory, anti-neuroinflammatory, and PTP1B inhibitory effects had not yet been investigated. Our results demonstrated that compounds **2**, **4**, and **5** inhibited LPS-induced overproduction of NO and PGE_2_ in BV2 microglial and RAW264.7 macrophage cells. Among these compounds, compound **2** was identified as the most active compound. Furthermore, compound **2** attenuated the mRNA expression of pro-inflammatory cytokines and the protein expression of iNOS and COX-2 in BV2 and RAW264.7 cells. Furthermore, the inhibitory effects of **2** were associated with the inactivation of NF-κB and p38 MAPK pathways. It is noteworthy that a minor modification of the dihydropyranobenzopyranone skeleton in compounds **1** and **2** produced a significant variation in the anti-inflammatory effect in the cellular models considered. This result suggests that this class of compound should be the subject of further investigation, particularly regarding structure and activity relationship. In addition, compounds **4** and **5**, which, respectively, are dihydropyranochromenone- and benzoylpyronone-type metabolites, have properties that suggest their pharmacologically evaluation for the treatment of diseases related to the regulation of PTP1B activity. According to the Michaelis –Menten kinetic model, compound **4** was identified as a noncompetitive inhibitor of the PTP1B enzyme, and compound **5** appeared to be a competitive inhibitor of this enzyme. Thus, this study demonstrates the potential value of marine-derived fungal isolates as a bioresource for bioactive compounds. The metabolites identified in this study should be further evaluated for their putative pharmacological properties.

## Figures and Tables

**Figure 1 marinedrugs-18-00247-f001:**
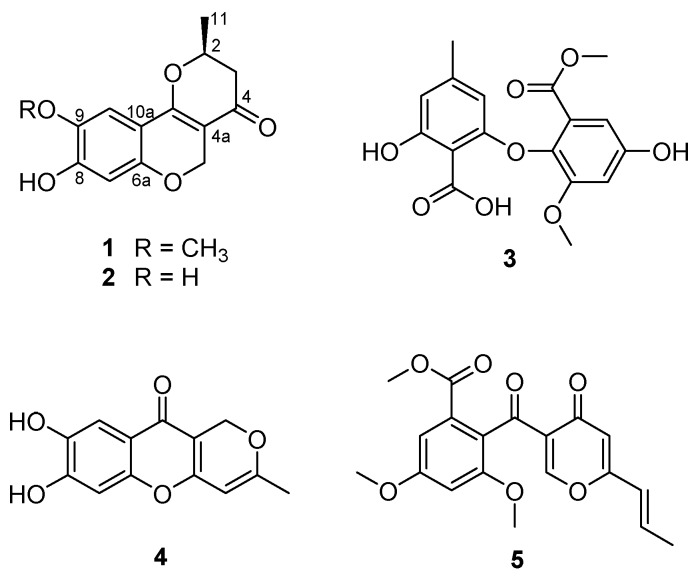
Chemical structures of compounds **1–5.**

**Figure 2 marinedrugs-18-00247-f002:**
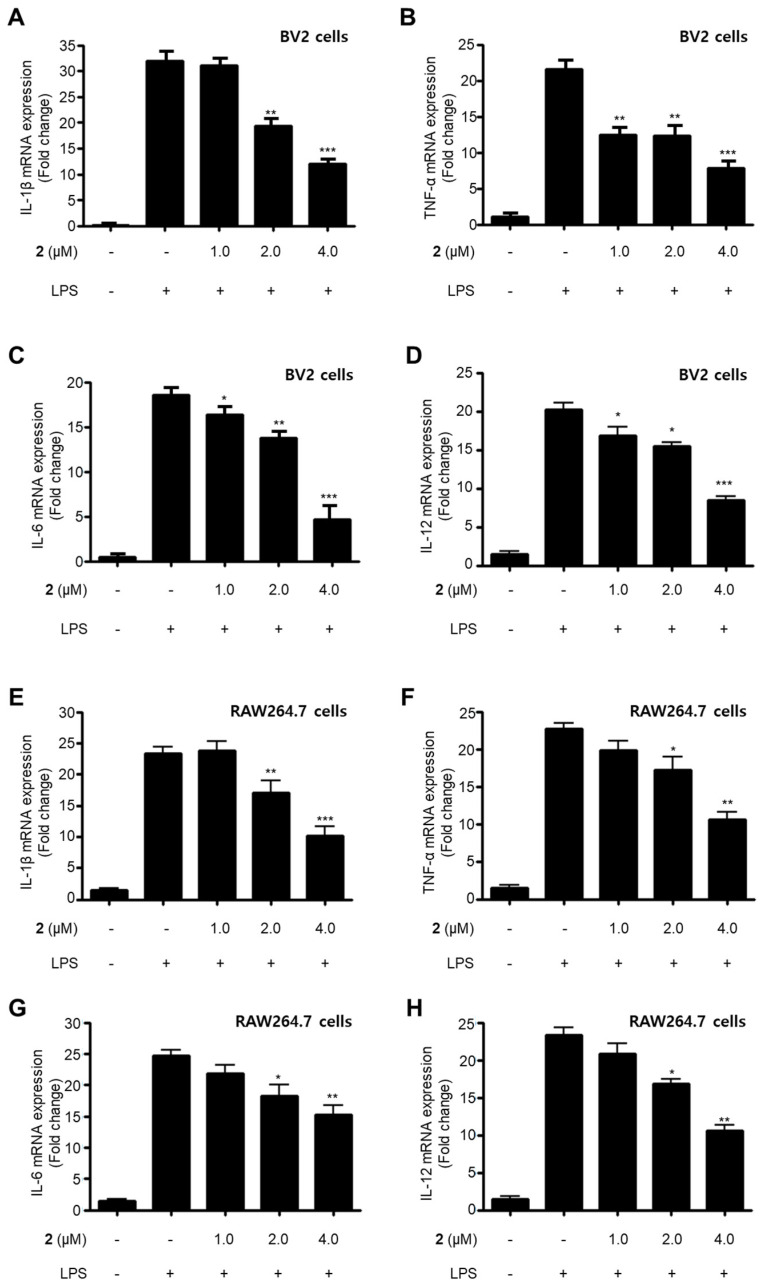
Suppressive effects of compound **2** on the mRNA expression of pro-inflammatory cytokines IL-1β (**A**,**E**), TNF-α (**B**,**F**), IL-6 (**C**,**G**), and IL-12 (**D**,**H**) in LPS-stimulated BV2 and RAW264.7 cells. After pre-treatment for 3 h with the indicated concentrations of compound **2**, the cells were stimulated for 12 h with LPS (1 μg/mL). Data represent the mean values of three experiments ± SD (* *p* < 0.05; ** *p* < 0.01; *** *p* < 0.001) compared to the LPS-treated group.

**Figure 3 marinedrugs-18-00247-f003:**
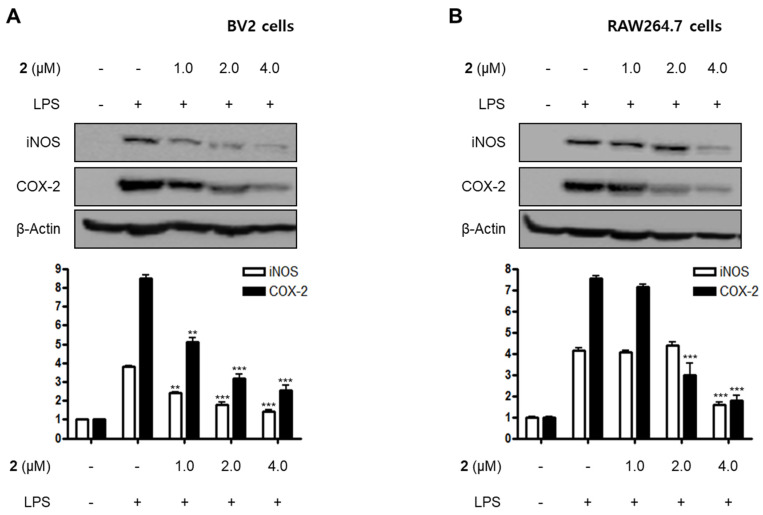
Effects of compound **2** on the protein expression of inducible nitric oxide synthase (iNOS) and cyclooxygenase-2 (COX-2) in LPS-induced BV2 (**A**) and RAW264.7 cells (**B**). Cells were pre-treated with compound **2** (1.0, 2.0, and 4.0 μM) for 3 h and stimulated with LPS (1 μg/mL) for 24 h. The representative blots of three independent analyses for iNOS and COX-2 expression are shown. Band intensity was quantified by densitometry and normalized to β-actin (** *p* < 0.01; *** *p* < 0.001) compared to the LPS-treated group.

**Figure 4 marinedrugs-18-00247-f004:**
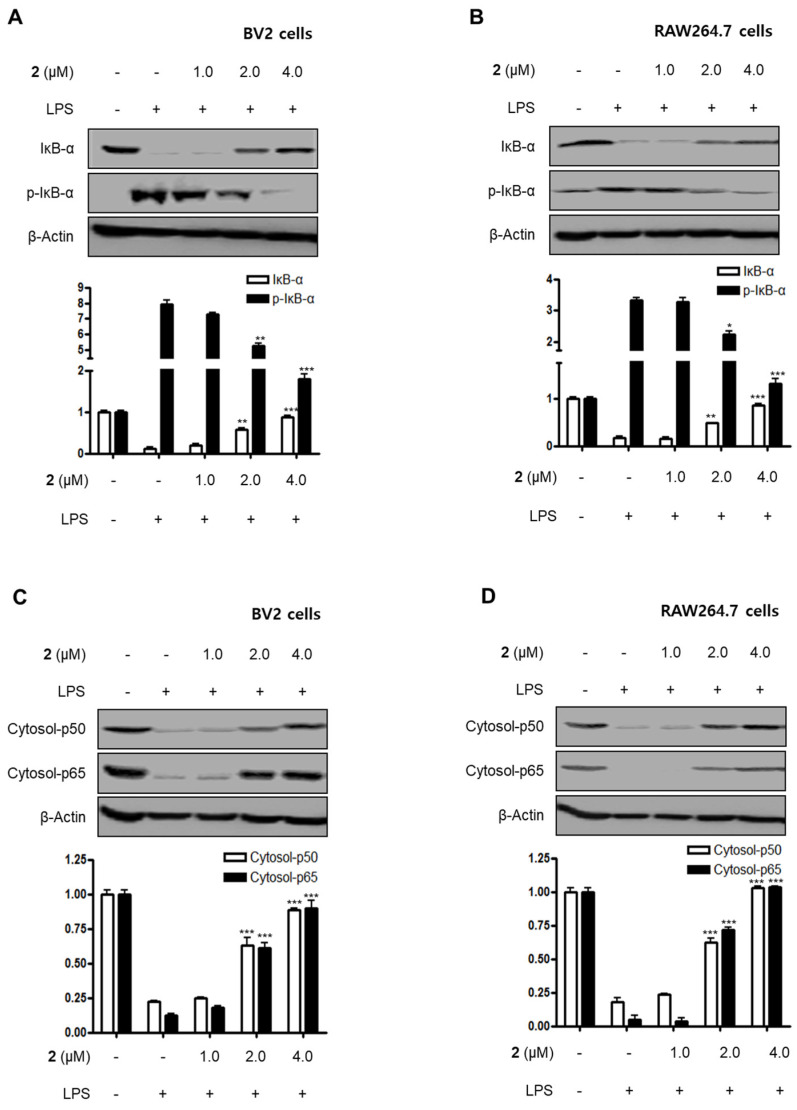
Effect of compound **2** on the activation of the NF-κB pathway in LPS-induced BV2 and RAW264.7 cells. After pre-treatment with compound **2** (1.0, 2.0, and 4.0 μM) for 3 h, the cells were stimulated with LPS for 1 h. (**A**–**F**) Proteins were obtained, and specific anti-IκB-α, anti-p- IκB-α, anti-p65, and anti-p50 antibodies were employed for western blot analysis. Representative blots of three independent experiments are shown. Band intensity was quantified by densitometry and normalized to β-actin for the cytoplasmic fraction and to proliferating cell nuclear antigen (PCNA) for the nuclear fraction. (**G, H**) NF-κB binding activity in the nuclear fraction was determined using an NF-κB ELISA kit by Active Motif (* *p* < 0.05; ** *p* < 0.01; *** *p* < 0.001) and then compared to that of the LPS-treated group.

**Figure 5 marinedrugs-18-00247-f005:**
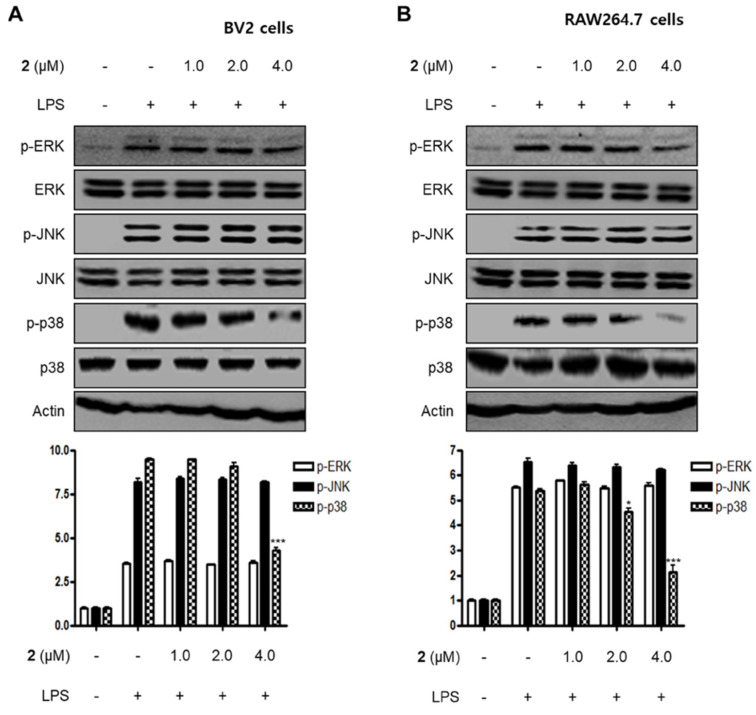
Effect of compound **2** on the activation of MAPK pathways in LPS-induced BV2 (**A**) and RAW264.7 cells (**B**). After pre-treatment with compound **2** (1.0, 2.0, and 4.0 μM) for 3 h, the cells were stimulated with LPS for 1 h. Western blot analysis was performed to determine the phosphorylated levels of extracellular signal-regulated kinases (ERK) (p-ERK), c-Jun N-terminal kinases (JNK) (p-JNK), and p38 MAPK (p-p38 MAPK). Representative blots of three independent experiments are shown. Band intensity was quantified by densitometry and normalized to β-actin. * *p* < 0.05; *** *p* < 0.001 as compared to the LPS-treated group.

**Figure 6 marinedrugs-18-00247-f006:**
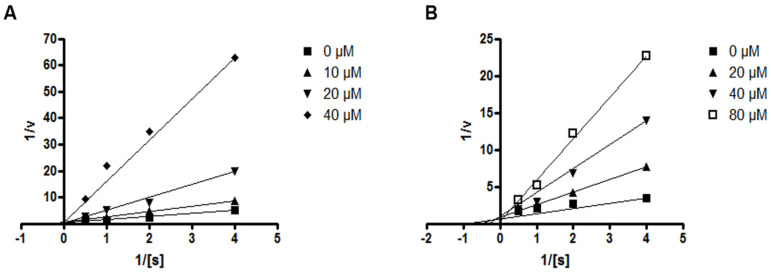
Lineweaver–Burk plots for compounds **4** (**A**) and **5** (**B**) representing the inhibition of PTP1B. The data represent the mean values ± SD of three experiments. The concentrations (μM) of compounds **4** and **5** are indicated.

**Table 1 marinedrugs-18-00247-t001:** NMR data of 9-*O*-methylneuchromenin (**1**).

Position	^δ^ _C_ ^a,b^	δ_H_ ^a,c^ (mult, *J* in Hz)	HMBC
2	77.3	4.72 (m)	
3	43.5	2.53 (m)	2, 4, 11
4	191.0	-	-
4a	102.4	-	-
5	64.1	5.03 (d, 12.0),4.73 (d, 12.0)	4, 4a, 6a, 10b
6a	156.3	-	-
7	104.8	6.34 (s)	6a, 8, 9, 10, 10b
8	154.5	-	-
9	144.8	-	-
10	108.9	7.06 (s)	6a, 8, 9, 10, 10b
10a	107.9	-	
10b	165.7	-	-
11	20.6	1.55 (d, 6.4)	2, 3, 4
12	57.0	3.83 (s)	9

^a^ Recorded in CD_3_OD, ^b^ 100 MHz, ^c^ 400 MHz.

**Table 2 marinedrugs-18-00247-t002:** Inhibitory effects (IC_50_ = μM) of compounds **1–5** on the overproduction of NO and prostaglandin E_2_ (PGE_2_) in lipopolysaccharide (LPS)-stimulated cells.

Compounds	NO Inhibitory Effects ^a^		PGE_2_ Inhibitory Effects ^a^	
	BV2 Cells	RAW264.7 Cells	BV2 Cells	RAW264.7 Cells
1	>80	>80	>80	50.2 ± 2.5
2	2.7 ± 0.1	4.7 ± 0.2	3.2 ± 0.2	4.1 ± 0.1
3	>80	>80	>80	53.4 ± 2.7
4	28.1 ± 1.4	41.5 ± 2.1	25.2 ± 1.3	30.0 ± 1.5
5	10.6 ± 0.5	40.1 ± 2.0	32.3 ± 1.6	>80

^a^ Values present mean ± SD of triplicate experiments.

**Table 3 marinedrugs-18-00247-t003:** Inhibitory effects of compounds **1****–5** on protein tyrosine phosphatase 1B (PTP1B).

Compounds	Inhibitory Effects on PTP1B ^a^
1	45.7% ^b^
2	37.8% ^b^
3	13.7% ^b^
4	19.2 ± 1.0
5	24.3 ± 1.2
Ursolic acid ^b^	3.1 ± 0.2

^a^ Values (IC_50_ = μM) represent mean ± SD of triplicate experiments; ^b^ inhibition percentage at 50 μM.

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
