# Peer review of "Anti-Inflammatory and Protein Tyrosine Phosphatase 1B Inhibitory Metabolites from the Antarctic Marine-Derived Fungal Strain Penicillium glabrum SF-7123"

_marinedrugs, 2020, doi:10.3390/md18050247_

Round 1

Reviewer 1 Report

This manuscript is an exhaustive in vitro study of the anti-inflammatory and tyrosine phosphatase 1B inhibitory properties of Antarctic marine fungal metabolites. The quality of the studies were impressive and clearly indicate that Compound 2 is a lead compound for further anti-inflammatory agent development. I hope that further in vitro SAR and in vivo efficacy studies are forthcoming. Congratulations on your efforts!

Author Response

Reviewer 1

This manuscript is an exhaustive in vitro study of the anti-inflammatory and tyrosine phosphatase 1B inhibitory properties of Antarctic marine fungal metabolites. The quality of the studies were impressive and clearly indicate that Compound 2 is a lead compound for further anti-inflammatory agent development. I hope that further in vitro SAR and in vivo efficacy studies are forthcoming. Congratulations on your efforts!

Response:  Thank you very much for your considerable comment.  We are looking forward to setting up further studies for SAR and/or in vivo efficacy of the lead compound.  We hope this answer would appropriately correspond to the comment. 

Reviewer 2 Report

The study describes the synthesis and characterization of chemical compounds from marine derived fungal strain. The authors analyzed the anti-inflammatory effect of the most active compound neuchromenin (2) and its potential mechanism of action using in vitro cell line model.

The analysis of the most active compound neuchromenin (2) is limited to only two mouse cell lines - one for microglial cells and another for mouse macrophage cell line. It would be highly informative to paralleled the information into human cell lines, and to test whether compound (2) is non-toxic to normal human cells at such a low concentration as in mouse cell lines.

Minor/specific concerns:

Abstract: Line 21-23. Please consider to separate the results on two different cell lines or modified the sentence for clarity.
Line 24-29: Please consider to revise/simplify the sentence. It's too long and appear full of jargons.

Please consider to add a conclusive sentence about the study importance.

Introduction: Line 68-71.
Please consider to revise this part in continuity. It appears to stand alone.

In figure 2, it would be important to add the panel for RAW264.7 cells analyzing the level of IL-1b, TNF-a, and IL-6 similar to BV2 cells, specifically, in view of results in Figure 3 for RAW264.7. Additionally, it appears that BV2 cells were 1st treated with (2) for 3 hours, and then with LPS. If so, please consider to change the figure labeling by moving (2) up and LPS down to maintain the consistency. Same applies to Figure 2 bottom part of the labeling.

Please consider to revise Line 137 & 138 as per results in Figure 3 Specific to RAW cells that attenuation was observed at 4.0 uM.

Table 2 appears inconsistent in terms of presentation (% vs IC50) and bit confusing with the word 'overproduction'.

LPS stimulation and incubation time period in Line 136 and Line #149-150 does not match. Please check for correction?

Author Response

Reviewer 2

The study describes the synthesis and characterization of chemical compounds from marine derived fungal strain. The authors analyzed the anti-inflammatory effect of the most active compound neuchromenin (2) and its potential mechanism of action using in vitro cell line model.

The analysis of the most active compound neuchromenin (2) is limited to only two mouse cell lines - one for microglial cells and another for mouse macrophage cell line. It would be highly informative to paralleled the information into human cell lines, and to test whether compound (2) is non-toxic to normal human cells at such a low concentration as in mouse cell lines.

Response:  Thanks for the comment.  We agree that our manuscript would be more informative if we could provide the effects of the isolated metabolites in human cell lines.  However, we thought that the evaluation of anti-inflammatory effects in two cellular models described in the current manuscript would be an important first step to report the informative biological effects of the metabolites because those cell lines have been frequently employed in research related to the anti-inflammatory agents.  Unfortunately, we do not have easy access to evaluate the effects of the metabolites in human cell lines, thus we would like to report the anti-inflammatory effects of the metabolites with the scope as described in the current manuscript.

Minor/specific concerns:

Abstract: Line 21-23. Please consider to separate the results on two different cell lines or modified the sentence for clarity.

Response:  Thanks for the comment.  We revised the manuscript according to the comment.  We described the effects for each cell line in separated sentences for clarity.  Please refer to lines 19-24 in the revised manuscript.

Line 24-29: Please consider to revise/simplify the sentence. It's too long and appear full of jargons. Please consider to add a conclusive sentence about the study importance.

Response:  Thanks for the comment.  We revised the manuscript according to the comment.  We segmented the sentence into short sentences for clear expression.  Please refer to lines 25-30 in the revised manuscript.

Please consider to add a conclusive sentence about the study importance.

Response:  Thanks for the comment.  We revised the manuscript according to the comment. We provided a sentence to point out the significance of the study.  Please refer to lines 34-35 in the revised manuscript.

Introduction: Line 68-71.
Please consider to revise this part in continuity. It appears to stand alone.

Response:  Thank you very much for your considerable comment. According to your comments, we revised the manuscript with rearranging some sentences and add a sentence stating the relationship between PTP1B and inflammation in microglia and macrophages. Please refer to lines 51-53 and lines 70-76 in the revised manuscript.

In figure 2, it would be important to add the panel for RAW264.7 cells analyzing the level of IL-1b, TNF-a, and IL-6 similar to BV2 cells, specifically, in view of results in Figure 3 for RAW264.7. Additionally, it appears that BV2 cells were 1st treated with (2) for 3 hours, and then with LPS. If so, please consider to change the figure labeling by moving (2) up and LPS down to maintain the consistency. Same applies to Figure 2 bottom part of the labeling.

Response:  Thank you very much for your considerable comment.  We added the inhibitory effects against upregulation of pro-inflammatory cytokines [tumor necrosis factor (TNF)-α, interleukin (IL)-1β, IL-6 and IL-12] mRNA expression in the LPS-treated RAW264.7 cells.  Please refer to lines 144-145 and Figure 2 in the revised manuscript.  In addition, we revised the Figures 2-5 according to the comment.  Please check the Figures 2-5 in the revised manuscript.

Please consider to revise Line 137 & 138 as per results in Figure 3 Specific to RAW cells that attenuation was observed at 4.0 uM.

Response:  Thanks for the comment.  We revised the manuscript according to the comment.  We have modified the sentence to correlate the result presented in the Figure 3.  Please refer to lines 145-148 in the revised manuscript.

Table 2 appears inconsistent in terms of presentation (% vs IC50) and bit confusing with the word 'overproduction'.

Response:  Thanks for the comment.  We revised the manuscript according to the comment.  We have changed the title and content of the table to avoid confusion.  Please refer to lines 152-154 in the revised manuscript.

LPS stimulation and incubation time period in Line 136 and Line #149-150 does not match. Please check for correction?

Response: Thank you very much for your considerable comment.  According to your comments, we revise the manuscript.  Please refer to line 159 in the revised manuscript.